

# ShExML: improving the usability of heterogeneous data mapping languages for first-time users

Herminio García-González[1], Iovka Boneva[2], Sławek Staworko[2], José Emilio Labra-Gayo[1] and Juan Manuel Cueva Lovelle[1]

[1] Deparment of Computer Science, University of Oviedo, Oviedo, Asturias, Spain
[2] University of Lille, INRIA, Lille, Nord-Pas-de-Calais, France

## ABSTRACT

Integration of heterogeneous data sources in a single representation is an active field with many different tools and techniques. In the case of text-based approaches—those that base the definition of the mappings and the integration on a DSL—there is a lack of usability studies. In this work we have conducted a usability experiment ($n = 17$) on three different languages: ShExML (our own language), YARRRML and SPARQL-Generate. Results show that ShExML users tend to perform better than those of YARRRML and SPARQL-Generate. This study sheds light on usability aspects of these languages design and remarks some aspects of improvement.

## INTRODUCTION

Data integration is the problem of mapping data from different sources so that they can be used through a single interface (*Halevy, 2001*). In particular, data exchange is the process of transforming source data to a target data model, so that it can be integrated in existing applications (*Fagin et al., 2005*). Modern data exchange solutions require from the user to define a *mapping* from the source data model to the target data model, which is then used by the system to perform the actual data transformation. This process is crucial to many applications nowadays as the number of heterogeneous data sources is growing (*Reinsel, Gantz & Rydning, 2018*).

Although many technologies have appeared through the years, the emergence of the semantic web (*Berners-Lee, Hendler & Lassila, 2001*) offered new perspectives for data integration. The semantic web principle recommends to represent entities through a unique Internationalized Resource Identifier (IRI) which allows creation of implicit links between distinct datasets simply by reusing existing IRIs. Moreover, the Resource Description Framework (RDF), which is the advocated data format for the semantic web, is compositional, meaning that one can simply fuse data sources without the use of a specific merger. These characteristics make RDF a privileged format for data integration and thus a target for data exchange and transformation.

Corresponding author
Herminio García-González, garciaherminio@uniovi.es

[1] see https://data.bnf.fr/en/about for more information on the project.

The most notable example of an RDF based data integration system is Wikidata (https://www.wikidata.org/) where multiple contributors—humans or robots—transform data from different sources and integrate it to the Wikidata data store. Another example is the data.bnf.fr[1] project that exposes in RDF format the catalog of the French National Library (BNF) by interlinking it with other datasets around the world.

Initially, the only way to perform these data transformations was to use ad-hoc scripts designed to take one data source and transform it to an RDF output. This supposed the creation of a dedicated script for every new input data source that needed to be converted. Such solutions are slow and costly to develop.

Later on, Domain Specific Language (DSL) approaches emerged which are able to define a translation in a declarative fashion instead of an imperative one. This technique lowers the development time, but a script for every different data source is still needed, which can be a maintenance issue.

More recent systems allow direct transformation of multiple data sources into a single representation. Some of them provide dedicated DSLs in which a single script defines the multi-source transformation, others provide graphical interfaces. This is an improvement compared to previous techniques as in principle it allows for faster development and improved maintainability (*Meester et al., 2019*). However, the adoption of such systems depends also on their *usability* (*Hanenberg, 2010*).

With usability in mind we have designed the ShExML (*García-González, Fernández-Álvarez & Gayo, 2018*) language that allows transformation and integration of data from XML and JSON sources in a single RDF output. ShExML uses Shape Expressions (ShEx) (*Prud'hommeaux, Labra Gayo & Solbrig, 2014*) for defining the desired structure of the output. ShExML has text based syntax (in contrast to graphical tools) and is intended for users that prefer this kind of representation. Our hypothesis is that for first-time users with some programming and Linked Data background, data integration is performed more easily using ShExML than using one of the existing alternatives. The consequent research questions that we study in the current paper are:

- RQ1: Is ShExML more usable for first-time users over other languages?
- RQ2: If true, can a relation be established between features support and usability for first-time users?
- RQ3: Which parts of ShExML—and of other languages—can be improved to increase usability?

In the case of this work we are going to focus on usability of tools based on a DSL and see how the design of the language can have an effect on usability and associated measures such as: development time, learning curve, etc.

The rest of the paper is structured as follows: 'Background' studies the related work, in 'Presentation of the Languages Under Study' the three languages are compared alongside a features comparison between them, in 'Methodology' we describe the methodology followed in the study, in 'Results' the results are presented along with their statistical analysis. In 'Discussion' we discuss and interpret the results and in 'Conclusions and Future Work' we draw some conclusions and propose some future lines from this work.

## BACKGROUND

We first review available tools and systems for generating RDF from different systems for data representation. These can be divided into one-to-one and many-to-one transformations. We also survey existing studies on the effectiveness of heterogeneous data mapping tools.

### One to one transformations

Much research work has been done in this topic where conversions and technologies were proposed to transform from a structured format (e.g., XML, JSON, CSV, Databases, etc.) to RDF.

#### From XML to RDF

In XML ecosystem many conversions and tools have been proposed:

*Miletic et al. (2007)* describe their experience with the transformation of RDF to XML (and vice versa) and from XML Schema to RDF Schema. *Deursen et al. (2008)* propose a transformation from XML to RDF which is based on an ontology and a mapping document. An approach to convert XML to RDF using XML Schema is reported by *Battle (2004)* and *Battle (2006)*. *Thuy et al. (2008)* describe how they perform a translation from XML to RDF using a matching between XML Schema and RDF Schema. The same procedure was firstly proved with a matching between DTD and RDF Schema by the same authors in (*Thuy et al., 2007*). *Breitling (2009)* reports a technique for the transformation between XML and RDF by means of the XSLT technology which is applied to astronomy data. Another approach that uses XSLT attached to schemata definitions is described by *Sperberg-McQueen & Miller (2004)*. However, use of XSLT for lifting purposes tends to end up in complex and non flexible stylesheets. Thus, *Bischof et al. (2012)* present XSPARQL, a framework that enables the transformation between XML and RDF by using XQuery and SPARQL to overcome the drawbacks of using XSLT for these transformations.

#### From JSON to RDF

Although in the JSON ecosystem there are less proposed conversions and tools, there are some works that should be mentioned.

*Müller et al. (2013)* present a transformation of a RESTful API serving interlinked JSON documents to RDF for sensor data. An RDF production methodology from JSON data tested on the Greek open data repository is presented by *Theocharis & Tsihrintzis (2016)*. *Freire, Freire & Souza (2017)* report a tool able to identify JSON metadata, align them with vocabulary and convert it to RDF; in addition, they identify the most appropriate entity type for the JSON objects.

#### From tabular form to RDF

The importance of CSV (along with its spreadsheet counterparts) has influenced work in this ecosystem:

*Ermilov, Auer & Stadler (2013)* present a mapping language whose processor is able to convert from tabular data to RDF. A tool for translating spreadsheets to RDF without the assumption of identical vocabulary per row is described by *Han et al. (2008)*. *Fiorelli et*

*al. (2015)* report a platform to import and lift from spreadsheet to RDF with a human-computer interface. Using SPARQL 1.1 syntax TARQL (http://tarql.github.io/) offers an engine to transform from CSV to RDF. CSVW proposed a W3C Recommendation to define CSV to RDF transformations using a dedicated DSL (*Tandy, Herman & Kellogg, 2015*).

### From databases to RDF

Along with the XML ecosystem, relational database transformation to RDF is another field:

*Bizer & Seaborne (2004)* present a platform to access relational databases as a virtual RDF store. A mechanism to directly map relational databases to RDF and OWL is described by *Sequeda, Arenas & Miranker (2012)*; this direct mapping produces a OWL ontology which is used as the basis for the mapping to RDF. Triplify (*Auer et al., 2009*) allows to publish relational data as Linked Data converting HTTP-URI requests to relational database queries. One of the most relevant proposals is R2RML (*Das, Sundara & Cyganiak, 2012*) that became a W3C Recommendation in 2012. R2RML offers a standard language to define conversions from relational databases to RDF. In order to offer a more intuitive way to declare mapping from databases to RDF, *Stadler et al. (2015)* presented SML which bases its mappings into SQL views and SPARQL construct queries.

More comprehensive reviews of tools and comparisons of tools for the purpose of lifting from relational databases to RDF are presented by (*Michel, Montagnat & Zucker, 2014*; *Hert, Reif & Gall, 2011*; *Sahoo et al., 2009*).

## Many to one transformations

Many to one transformations is a recent topic which has evolved to overcome the problem that one to one transformations need a different solution for each format and that subsequently must be maintained.

### Source-centric approaches

Source-centric approaches are those that, even giving the possibility of transforming multiple data sources to multiple serialisation formats, they base their transformation mechanism in one to one transformations. This can deliver optimal results—if exported to RDF—due to RDF compositional property. Some of the tools available are: OpenRefine (http://openrefine.org/) which allows to perform data cleanup and transformation to other formats, DataTank (http://thedatatank.com/) which offers transformation of data by means of a RESTful architecture, Virtuoso Sponger (http://vos.openlinksw.com/owiki/wiki/VOS/VirtSponger) is a middleware component of Virtuoso able to transform from a data input format to another serialisation format, RDFizers (http://wiki.opensemanticframework.org/index.php/RDFizers) employs the Open Semantic Framework to offer hundreds of different format converters to RDF. The Datalift (*Scharffe et al., 2012*) framework also offers the possibility of transforming raw data to semantic interlinked data sources.

### Text-based approaches

The use of a mapping language as the way to define all the mappings for various data sources was first introduced by RML (*Dimou et al., 2014*) which extends R2RML syntax (Turtle

based) to cover heterogeneous data sources. With RML implementations it is possible to gather data from: XML, JSON, CSV, Databases and so on; and put them together in the same RDF output. A similar approach was also followed in KR2RML (*Slepicka et al., 2015*) which proposed an alternative interpretation of R2RML rules paired with a source-agnostic processor facilitating data cleaning and transformation. To deal with non-relational databases, *Michel et al. (2015)* presented xR2RML language which extends R2RML and RML specifications. Then, SPARQL-Generate (*Lefrançois, Zimmermann & Bakerally, 2016*) was proposed which extends SPARQL syntax to serve as a mapping language for heterogeneous data. This solution has the advantage of using a very well-known syntax in the semantic web community and that its implementation is more efficient than RML main one (i.e., RMLMapper (https://github.com/RMLio/RML-Mapper)) (*Lefrançois, Zimmermann & Bakerally, 2017*). To offer a simpler solution for users of text-based approaches, YARRRML (*Heyvaert et al., 2018*) was introduced which offers a YAML based syntax and its processor (https://github.com/RMLio/yarrrml-parser) performs a translation to RML rules.

### Graphical-based approaches

Graphical tools offer an easier way to interact with the mapping engine and are more accessible to non-expert users. Some of the tools mentioned in the previous source-centric approaches section have graphical interfaces, like OpenRefine and DataTank. RMLEditor (*Heyvaert et al., 2016*) offers a graphical interface for the creation of RML rules.

### Related studies

Some studies have been made to evaluate available tools and languages. *Lefrançois, Zimmermann & Bakerally (2017)* compared SPARQL-Generate implementation to RMLMapper. Their results showed that SPARQL-Generate has a better computational performance when transforming more than 1500 CSV rows in comparison with RMLMapper. They also concluded that SPARQL-Generate language is easier to learn and use for semantic web practitioners (who are likely already familiar with SPARQL), but this was based on a limited analysis of the cognitive complexity of query/mappings in the two languages. RMLEditor, a graphical tool to generate RML rules was proposed by *Heyvaert et al. (2016)*. They performed a usability evaluation for their tool with semantic web experts and non-experts. In the case of semantic web experts they also evaluate the differences between the textual approach (RML) and this new visual one. However, RMLEditor was neither compared with other similar tools nor RML with other languages. *Heyvaert et al. (2018)* proposed YARRRML as a human-readable text-based representation which offers an easier layer on top of RML and R2RML. However, the authors did not present any evaluation of this language. *Meester et al. (2019)* made a comparative characteristic analysis of different mapping languages. However, a qualitative analysis is not performed and usability is only mentioned in NF1 "Easy to use by Semantic Web experts" which only YARRRML and SPARQL-Generate achieve.

Thus, to the best of our knowledge no usability study was performed in these languages which share the easiness of use as one of their goals. Therefore, we introduce this study as a first step into the usability evaluation of heterogeneous data mapping languages.

## PRESENTATION OF THE LANGUAGES UNDER STUDY

In this section we compare YARRRML, SPARQL-Generate and ShExML syntax by means of a simple example. These three tools each offer a DSL able to define mappings for heterogeneous data sources like we have seen in the previous section and their designers share the goal to be user friendly (*Meester et al., 2019*; *García-González, Fernández-Álvarez & Gayo, 2018*). RML and similar alternatives are not included in the comparison because they have a verbose syntax very close to the RDF data model. While it might be an interesting solution for users without any programming knowledge but familiar with RDF, we consider it more like a lower level middle language to compile to rather than a language to be used by programmers and data engineers. Indeed, YARRRML and ShExML engines are able to compile their mappings to RML.

For the sake of the example two small files on JSON and XML are presented in Listing 1 and Listing 2 respectively. Each one of these files define two films with 6 attributes—that could differ on name and structure—that will be translated to the RDF output showed in Listing 3. In this example, and with the aim to keep it simple, different ids are used in each entity; however, it is possible to use objects with same ids that could be merged into a single entity or divided into different new entities depending on users' intention.

Listing 1: JSON films file

```
{
    "films": [
        {
            "id": 3,
            "title": "Inception",
            "date": "2010",
            "countryOfOrigin": "USA",
            "director": "Christopher Nolan",
            "screenwriter": "Christopher Nolan"
        },
        {
            "id": 4,
            "title": "The Prestige",
            "date": "2006",
            "countryOfOrigin": "USA",
            "director": "Christopher Nolan",
            "screenwriter": ["Christopher Nolan",
              "Jonathan Nolan"]
        }
    ]
}
```

Listing 2: XML films file

```
<films>
    <film id="1">
        <name>Dunkirk</name>
        <year>2017</year>
```

```
            <country>USA</country>
            <director>Christopher Nolan</director>
            <screenwriters>
                <screenwriter>Christopher Nolan</screenwriter>
            </screenwriters>
        </film>
        <film id="2">
            <name>Interstellar</name>
            <year>2014</year>
            <country>USA</country>
            <director>Christopher Nolan</director>
            <screenwriters>
                <screenwriter>Christopher Nolan</screenwriter>
                <screenwriter>Jonathan Nolan</screenwriter>
            </screenwriters>
        </film>
</films>
```

Listing 3: RDF output

```
@prefix :       <http://example.com/> .

:4      :country         "USA" ;
        :screenwriter    "Jonathan Nolan" ,
            "Christopher Nolan" ;
        :director        "Christopher Nolan" ;
        :name            "The Prestige" ;
        :year            :2006 .

:3      :country         "USA" ;
        :screenwriter    "Christopher Nolan" ;
        :director        "Christopher Nolan" ;
        :name            "Inception" ;
        :year            :2010 .

:2      :country         "USA" ;
        :screenwriter    "Jonathan Nolan" ,
            "Christopher Nolan" ;
        :director        "Christopher Nolan" ;
        :name            "Interstellar" ;
        :year            :2014 .

:1      :country         "USA" ;
        :screenwriter    "Christopher Nolan" ;
        :director        "Christopher Nolan" ;
        :name            "Dunkirk" ;
        :year            :2017 .
```

## YARRRML

Listing 4: YARRRML transformation script for the films example

```
prefixes:
  ex: "http://example.com/"

mappings:
  films_json:
    sources:
      - ['films.json~jsonpath', '$.films[*]']
    s: ex:$(id)
    po:
      - [ex:name, $(title)]
```

```
        - [ex:year, ex:$(date)~iri]
        - [ex:director, $(director)]
        - [ex:screenwriter, $(screenwriter)]
        - [ex:country, $(countryOfOrigin)]
  films_xml:
    sources:
      - ['films.xml~xpath', '//film']
    s: ex:$(@id)
    po:
      - [ex:name, $(name)]
      - [ex:year, ex:$(year)~iri]
      - [ex:director, $(director)]
      - [ex:screenwriter, $(screenwriters/screenwriter)]
      - [ex:country, $(country)]
```

YARRRML is designed with human-readability in mind which is achieved through a YAML based syntax. Listing 4 shows the mappings `films_json` and `films_xml` for our films example. Each mapping starts with a source definition that contains the query to be used as iterator, e.g., `//film`. It is followed by the definition of the output given by a subject definition (`s:`) and a number of associated predicate-object definitions (`po:`). Subject and predicate-object definitions can use "partial" queries relative to the iterator to populate the subject and object values. This way of defining mappings is very close to RML; YARRRML actually does not provide an execution engine but is translated to RML.

## SPARQL-Generate

Listing 5: SPARQL-Generate transformation script for the films example

```
BASE <http://example.com/>
PREFIX iter: <http://w3id.org/sparql-generate/iter/>
PREFIX fun: <http://w3id.org/sparql-generate/fn/>
PREFIX rdfs: <http://www.w3.org/2000/01/rdf-schema#>
PREFIX xsd: <http://www.w3.org/2001/XMLSchema#>
PREFIX : <http://example.com/>
PREFIX dbr: <http://dbpedia.org/resource/>
PREFIX schema: <http://schema.org/>
PREFIX sc: <http://purl.org/science/owl/sciencecommons/>

GENERATE {
  ?id_json :name ?name_json ;
       :year ?year_json ;
       :director ?director_json ;
       :country ?country_json .

  GENERATE {
    ?id_json :screenwriter ?screenwriter_json .
  }
  ITERATOR iter:Split(?screenwriters_json, ",")
    AS ?screenwriters_json_iterator
  WHERE {
    BIND(REPLACE(?screenwriters_json_iterator,
      "\\[|\\]|\"", "")
    AS ?screenwriter_json)
  } .

  ?id_xml :name ?name_xml ;
       :year ?year_xml ;
       :director ?director_xml ;
       :country ?country_xml .
```

```
  GENERATE {
    ?id_xml :screenwriter ?screenwriter_xml .
  }
  ITERATOR iter:XPath(?film_xml,
    "/film/screenwriters[*]/screenwriter")
    AS ?screenwriters_xml_iterator
  WHERE {
    BIND(fun:XPath(?screenwriters_xml_iterator,
    "/screenwriter/text()") AS ?screenwriter_xml)
  } .

}
ITERATOR iter:JSONPath(
 <https://raw.githubusercontent.com/herminiogg/ShExML/
 master/src/test/resources/filmsPaper.json>,
 "$.films[*]") AS ?film_json
ITERATOR iter:XPath(
 <https://raw.githubusercontent.com/herminiogg/ShExML/
 master/src/test/resources/filmsPaper.xml>,
 "//film") AS ?film_xml
WHERE {
 BIND(IRI(CONCAT("http://example.com/",
   STR(fun:JSONPath(?film_json,"$.id")))) AS ?id_json)
 BIND(fun:JSONPath(?film_json, "$.title") AS ?name_json)
 BIND(fun:JSONPath(?film_json, "$.director")
   AS ?director_json)
 BIND(IRI(CONCAT("http://example.com/",
   fun:JSONPath(?film_json, "$.date"))) AS ?year_json)
 BIND(fun:JSONPath(?film_json, "$.countryOfOrigin")
   AS ?country_json)
 BIND(fun:JSONPath(?film_json, "$.director")
   AS ?directors_json)
 BIND(fun:JSONPath(?film_json, "$.screenwriter")
   AS ?screenwriters_json)
 BIND(IRI(CONCAT("http://example.com/",
   fun:XPath(?film_xml,"/film/@id"))) AS ?id_xml)
 BIND(fun:XPath(?film_xml, "/film/name/text()")
   AS ?name_xml)
 BIND(fun:XPath(?film_xml, "/film/director/text()")
   AS ?director_xml)
 BIND(IRI(CONCAT("http://example.com/",
   fun:XPath(?film_xml, "/film/year/text()")))
     AS ?year_xml)
 BIND(fun:XPath(?film_xml, "/film/country/text()")
   AS ?country_xml)
}
```

SPARQL-Generate is an extension of SPARQL 1.1 for querying heterogeneous data sources and creating RDF and text. It offers a set of SPARQL binding functions and SPARQL iterator functions to achieve this goal. The mapping for our films example is shown in Listing 5. The output of the mapping is given within the GENERATE clauses and can use variables and IRIs, while queries, IRI and variable declarations are declared in the WHERE clause. SPARQL-Generate is an expressive language that can be further extended using the SPARQL 1.1 extension system. On the other side, SPARQL-Generate scripts tend to be verbose compared to the other two languages studied in this paper.

## ShExML

Listing 6: ShExML transformation script for the films example

```
PREFIX : <http://example.com/>
SOURCE films_xml_file <
  https://raw.githubusercontent.com/herminiogg/
  ShExML/master/src/test/resources/filmsPaper.xml>
SOURCE films_json_file <
  https://raw.githubusercontent.com/herminiogg/
  ShExML/master/src/test/resources/filmsPaper.json>
ITERATOR film_xml <xpath: //film> {
    FIELD id <@id>
    FIELD name <name>
    FIELD year <year>
    FIELD country <country>
    FIELD director <director>
    FIELD screenwriters <screenwriters/screenwriter>
}
ITERATOR film_json <jsonpath: $.films[*]> {
    FIELD id <id>
    FIELD name <title>
    FIELD year <date>
    FIELD country <countryOfOrigin>
    FIELD director <director>
    FIELD screenwriters <screenwriter>
}
EXPRESSION films <films_xml_file.film_xml
  UNION films_json_file.film_json>

:Films :[films.id] {
    :name [films.name] ;
    :year :[films.year] ;
    :country [films.country] ;
    :director [films.director] ;
    :screenwriter [films.screenwriters] ;
}
```

ShExML, our proposed language, can be used to map XML and JSON documents to RDF. The ShExML mapping for the films example is presented in Listing 6. It consists of source definitions followed by iterator definitions. The latter define structured objects which fields are populated with the results of source queries. The output of the mapping is described using a Shape Expression (ShEx) (*Prud'hommeaux, Labra Gayo & Solbrig, 2014*; *Boneva, Labra Gayo & Prud'hommeaux, 2017*) which can refer to the previously defined fields. The originality of ShExML, compared to the other two languages studied here, is that the output is defined only once even when several sources are used. This is a design choice that allows the user to separate concerns: how to structure the output on the one hand, and how to extract the data on the other hand.

## Comparing languages features

In this subsection we compare languages features and what operations are supported or not in each language (see Table 1).

Iterators, sources, fields, unions and so on are common to the three languages as they have the same objective. They have different syntaxes, as it can be seen in the three examples, but from a functionality point of view there are no differences.

**Table 1  Features comparison between the three languages.**

| Features | | ShExML | YARRRML | SPARQL-Generate |
|---|---|---|---|---|
| **Source and output definition** | **Defining output** | Shape expression | Subject and predicate-object definitions | Generate clause |
| | **IRIs generation** | Prefix and value generation expression (concatenation) | Prefix and value generation expression (array) | Variable (previous use of concat function) or string interpolation |
| | **Datatypes & Language tags** | Yes | Yes | Yes |
| **Multiple results from a query** | | Treated like an array | Treated like an array | Need to iterate over the results |
| **Transformations** | | Limited (Matchers and String operators). | FnO hub | Functions for strings and extension mechanism |
| **Output formats** | **Output** | RDF | RDF | RDF and any text-based format |
| | **Translation** | RML | RML | No translation offered |
| **Link between mappings** | | Shape Linking and JOIN keyword (do not fully cover YARRRML feature) | Yes (conditions allowed) | Nested generate clauses, filter clauses and extension mechanism |
| **Conditional mapping generation** | | No | Yes (Function and conditional clause) | Yes (Filter clause and extension mechanism) |

**Source and output definition and their artefacts:** As we saw, the mechanism to define the form of the RDF output has different flavour in the three languages: subject and predicate-object definitions for every source in YARRRML; GENERATE clauses for every source in SPARQL-Generate; a single Shape Expression in ShExML. Additionally, the three languages offer slightly different operators for constructing the output values. All of them typically obtain IRIs by concatenating a source value to some prefix, and reuse literal values as is. YARRRML supports the generation of multiple named graphs whereas SPARQL-Generate can only generate one named graph at a time and ShExML only generates RDF datasets.

**Multiple results:** The handling of multiple results, like it occurs on the screenwriters case, is different between SPARQL-Generate and the two other languages. In YARRRML and ShExML if a query returns multiple results they are treated like a list of them. However, in SPARQL-Generate this functionality must be explicitly declared like it can be seen in Listing 5. It leads to complex iterator definitions like the one used in JSON screenwriters one.

**Transformations:** The possibility of transforming the output to another value by means of a function is something very useful for different purposes when building a knowledge graph. Therefore, in YARRRML this is supported through the FnO mechanism (*Meester et al., 2017*) which offers a way to define functions inside mapping languages in a declarative fashion. SPARQL-Generate offers some functions for strings embedded inside the SPARQL binding functions mechanism; however, it is possible to extend the language through the SPARQL 1.1 extension mechanism. In the case of ShExML, only Matchers and String operations are offered for transformation purposes.

**Other formats output:** Output format on YARRRML and ShExML is limited to RDF; whereas, in SPARQL-Generate it is possible to also generate plain text, enabling the potential transformation to a lot of different formats. In this aspect, SPARQL-Generate presents a much more flexible output. Converserly, YARRRML and ShExML engines offer a translation of their mappings to RML rules which improves interoperability with other solutions.

**Link to other mappings:** In YARRRML there is the possibility to link mappings between them. This functionality is provided by giving the name of the mapping to be linked and the condition that must be satisfied (e.g., ID of mapping A equal to ID of mapping B). This can be useful when the subject is generated with a certain attribute but this attribute does not appear on the other file so the linking should be done using another attribute. In ShExML this can be partially achieved by Shape linking—which is a syntactic sugar to avoid repeating an expression twice—and by the Join clause which gives an implementation for primary interlinking covering a subset of what is covered with YARRRML mapping linking. In SPARQL-Generate this can be achieved using nested Generate clauses and Filter clauses.

**Conditional mapping generation:** Sometimes there is the need to generate triples only in the case that some condition is fulfilled. In YARRRML this is achieved using the conditional clause and a function. In SPARQL-Generate this can be obtained with the SPARQL 1.1 Filter clauses and also with the extensibility mechanism offered by the language. In ShExML this is not possible currently.

**Further features of SPARQL-Generate:** Apart from what has been presented in the previous point, SPARQL-Generate, as being based on SPARQL 1.1, offers more expressiveness than the other two languages. One possibility that emerges from that is the use of defined variables. For example, it is possible to define an iterator of numbers and then use that numbers to request different parts of an API. This versatility enables the creation of very complex and rich scripts that can cover a lot of use cases. It is natural to expect that learning to use the full capabilities of SPARQL-Generate is complex, as the language offers a lot of features. In our experiments, however, only some basic features of the language were required and, as is shown in 'Results', it appears that SPARQL-Generate design did not help test subjects to solve the proposed tasks easily.

## METHODOLOGY

In order to test our hypothesis that ShExML is easier for first-time users only experienced in programming and the basics of linked data, an experiment was carried out. The University of Oviedo granted ethical approval to carry out the described study. Verbal consent was requested before starting the experiment.

### Experiment design

The selected tools were YARRRML (http://rml.io/yarrrml/), SPARQL-Generate (https://ci.mines-stetienne.fr/sparql-generate/) and ShExML (http://shexml.herminiogarcia.com/). We decided not to include RML (http://rml.io/) and similar alternatives for the same reason mentioned on 'Presentation of the Languages Under Study'. Three manuals were designed for the students based on the example about films that described how the integration can be
[2]Material can be consulted on: https://github.com/herminiogg/shexml-paper-2019-data/tree/master/experiment-material.

done with each tool.[2] The experiment was designed to be performed in each tool dedicated online environment, which are available through the Internet as a webpage.

In addition, a small manual was developed to guide the students along the experiment and to inform them about the input files and which are the expected outputs[2]. This manual contained two tasks to perform during the experiment which were designed to be performed sequentially, i.e., the student should finish the first task before starting with the second one. The first task was the mapping and integration of two files (JSON and XML) with information about books which should be mapped in a unique RDF graph. The final output should be equal to the one given in the guide. The second task was to modify the script done in the previous task so that the prices are separated and can be compared between markets. In other words, that multiple prices are tagged individually referring to the market where the specific price was found, like they were in the input files. This second task gives us an intuition on how easy is to modify an existing set of data mapping rules in each language.

The study was designed as a mixed method approach, including a quantitative analysis and a qualitative analysis. For the quantitative analysis measures, Mousotron (http://www.blacksunsoftware.com/mousotron.html) was used which allows to register the number of keystrokes, the distance travelled by the mouse and so on. For the qualitative analysis two Office 365 forms were used with questions based on a Likert scale (see questions in Table 2). In addition, the elapsed time was calculated from timestamps in the Office 365 forms.

## Conduction

The sample consisted on 20 students (four women and 13 men) of the MSc in Web Engineering first-year course (out of two years) at the University of Oviedo (http://miw.uniovi.es/). Most of them have a bachelor degree (240 ECTS credits) in computer science or similar fields. They were receiving a semantic web course of two weeks—a total of 30 hours (3 hours per day)—where they were introduced to semantic technologies like: RDF, SPARQL, ShEx, etc. Before this course they had not previous knowledge on semantic web technologies. Regarding prior knowledge of YAML by subjects, even though it is normally known and used by developers, we could not assure it. The experiment was hosted the final day of the course.

The experiment was conducted in their usual classroom and with their whole-year-assigned computers. So that they were in a confortable environment and with a computer they are familiar with. The three tools were assigned to the students in a random manner. Each student received the printed manual for its assigned tool and they were given a time of 20 minutes to read it, test the language in the online environment, and ask doubts and questions. Once these 20 minutes were elapsed the printed experiment guide was given to the students and they were explained about the experiment proceeding with indications about Mousotron operation.

In particular the procedure followed to perform the whole experiment was:

1. Open the assigned tool on the dedicated webpage and clear the given example.
2. Open Mousotron and reset it.

**Table 2 Statements to evaluate by the students based on a 5 point Likert scale.**

| Questionnaire | Statement | Obtained Variable |
| --- | --- | --- |
| 1 | The experience with the tool was satisfactory | General satisfaction level |
| 1 | The tool was easy to use | Easiness of use |
| 1 | The mapping definitions was easy | Mapping definition easiness |
| 1 | The language was easy to learn | Learnability |
| 1 | I find that these tool can be useful in my work | Applicability |
| 1 | The coding in this tool was intuitive | Intuitiveness |
| 1 | The language design leads to commit some errors | Error proneness |
| 1 | The error messages were useful to solve the problems | Error reporting usefulness |
| 2 | It was easy to define different predicates for the price | Modifiability |

3. Proceed with task 1 (start time registered for elapsed time calculation).
4. Once task 1 is finished, capture Mousotron results (screenshot) and fill the first Office 365 questionnaire.
5. Reset Mousotron and proceed with task 2.
6. Once task 2 is finished, capture Mousotron results (screenshot) and fill the second Office 365 questionnaire.

*Analysis*

The quantitative results were dump into an Excel sheet and anonymised. Although many results can be used as given by the students, some of them need to be calculated. This is the case of elapsed time (on both tasks), completeness percentage and precision. Elapsed time in the first task ($t_{t1}$) was calculated as the subtraction of questionnaire 1 beginning time ($st_{q1}$) and experiment start time ($st_e$), i.e., ($t_{t1} = st_{q1} - st_e$). Elapsed time in the second task ($t_{t2}$) was calculated as the subtraction of questionnaire 1 ending time ($et_{q1}$) and questionnaire 2 beginning time ($st_{q2}$), i.e., ($t_{t2} = st_{q2} - et_{q1}$).

Completeness percentage was calculated from three measures: the proportion of correctly generated triples contributed 50%, the proportion of data correctly translated contributed 25% and the proportion of correctly generated prefixes and datatypes as a 25%. This design gives more importance to the structure, which is the main goal when using these tools. Other aspects, like correct data (i.e., the object part of a triple), prefixes (i.e., using the correct predicate for the subject, the predicate and the object in case of an IRI) and the datatype (i.e., putting the correct xsd type in case of a literal object) are a little less valued as these errors could come more easily from a distraction or an oversight. Let *CP* be the completeness percentage, $t$ the number of triples, $d$ the number of data gaps and $p\&dt$ the number of prefixes and datatypes, so the calculation of the completeness percentage can be expressed as:

$$CP = 0.5 * \frac{t_{total} - t_{generated}}{t_{total}} + 0.25 * \frac{d_{total} - d_{generated}}{d_{total}} + 0.25 * \frac{p\&dt_{total} - p\&dt_{generated}}{p\&dt_{total}}.$$

Finally, precision was calculated as the division of current student elapsed time by minimum elapsed time of all students, multiplied by the completeness percentage. This precision formulation gives us an intuition on how fast was some student in comparison

with the fastest student and with a correction depending on how well his/her solution was. Let $t_{sn}$ be the elapsed time of student $n$ and $CP_{sn}$ the completeness percentage of student $n$ calculated with the previous formula.

$$Precision_{sn} = \frac{t_{sn}}{min(\{t_{s1}, ..., t_{sn}\})} * CP_{sn}.$$

The results of the qualitative analysis were only anonymised as they can be directly used from the Office 365 output.

For the analysis the IBM SPSS version 24 was used. We planned a One Way ANOVA test within the three groups in the quantitative analysis where a normal distribution was found and the Kruskal-Wallis test where not. The qualitative analysis comparison between three groups was established using the Kruskal-Wallis test. The report and analysis of the results was made using *Field (2013)* as guidance and using the suggested APA style as a standard manner to report statistical results.

## Threat to validity

In this experiment we have identified the following threats to its validity.

### Internal validity

We have identified the following internal validity threats in the experiment design:

- More expertise in some specific tool: In semantic web area—as in other areas—people tend to be more expert in some specific technologies and languages. The derived risk is that this expertise can have an influence on final results. To alleviate this we have selected MSc students that are studying the same introductory semantic web course and we have assigned the tools in a random manner.
- Not homogeneous group: It is possible that the selected group is not homogeneous on skills and previous knowledge. To mitigate this we have applied the same measures as for the previous threat: Students of a semantic web course and a randomised tool assignment.
- Unfamiliar environment: In usability studies, unfamiliar environments can play a role on final conclusions. Therefore, we opted to run the experiment in a well-known environment for the students, that is, their whole-year classroom.
- More guide and information about one tool: As we have designed one of the languages, it could lead to a bias in information delivery. To try to mitigate this threat we developed three identical manuals for each tool. Questions and doubts were answered equally for all the students and tools.

### External validity

Following the measures taken in the internal validity threats we identified the corresponding external validity ones:

- Very focused sample: As we have restricted the profile of the sample to students of a MSc course which are more or less within the same knowledge level, there is the risk that these findings cannot be extrapolated for other samples or populations. It is possible that for semantic web practitioners—with different interests and expertises—these findings
[3] Original datasets available on: https://github.com/herminiogg/shexml-paper-2019-data/tree/master/datasets.

are not applicable. However, the intention of this study was to evaluate usability with first-time users as a first step to guide future studies.

## RESULTS

From the 20 students of the sample,[3] in the first task, three of them left the experiment without making any questionnaire, two for SPARQL-Generate and one for YARRRML. In the second task, only seven out of the 20 students made the questionnaire, six for ShExML and 1 for YARRRML. The statistical analysis was made using the IBM SPSS software, version 24.

**Task 1:** As previously stated, the number of students that finished—correctly or not—the proposed task was 17. Descriptive statistics can be seen in Table 3. Comparison of three groups was made by means of a One Way ANOVA which results showed significant differences on elapsed seconds $F(2, 14) = 6.00$, $p = .013$, $\omega = .60$. As completeness percentage and precision are not following a normal distribution on SPARQL-Generate group ($W(4) = .63$, $p = .001$ and $W(4) = .63$, $p = .001$), the comparison was established by means of the Kruskal-Wallis test which showed significant differences in both variables ($H(2) = 9.73$, $p = .008$ and $H(2) = 9.68$, $p = .008$). Post hoc test for elapsed seconds using the Gabriel's criterion showed significant differences between ShExML group and YARRRML group ($p = .016$). Post hoc test for completeness percentage and precision using the Bonferroni's criterion showed significant differences between ShExML and SPARQL-Generate ($p = .012$, $r = .87$ and $p = .012$, $r = .87$). Likert scale questionnaire results ($\alpha = 0,73$) (see Fig. 1) were analysed using Kruskal-Wallis test which resulted in significant differences between groups for variables general satisfaction level ($H(2) = 6.28$, $p = .043$), easiness of use ($H(2) = 9.82$, $p = .007$), mapping definition easiness ($H(2) = 10.25$, $p = .006$) and learnability ($H(2) = 8.63$, $p = .013$). Bonferroni's criterion was used as post hoc test for the variables with significant differences. For general satisfaction level significant differences were found between ShExML and YARRRML ($p = .039$, $r = .69$). For easiness of use significant differences were found between ShExML and YARRRML ($p = .011$, $r = .81$). For mapping definition easiness significant differences were found between ShExML and SPARQL-Generate ($p = .013$, $r = .90$) and between ShExML and YARRRML ($p = .037$, $r = .69$). For learnability significant differences were found between ShExML and SPARQL-Generate ($p = .042$, $r = .78$) and between ShExML and YARRRML ($p = .040$, $r = .69$).

**Task 2:** In this task only seven students reached this step: 6 for ShExML and 1 for YARRRML. Descriptive statistics of this task can be seen in Table 4. No significant differences were found in any of the variables. In subjective variable analysis (see Fig. 2) no significant differences were found.

## DISCUSSION

### Statistical results discussion

Results of task 1 show that variables like keystrokes, left button clicks, right button clicks, mouse wheel scroll and meters travelled by the mouse, do not have a significant

**Table 3** Descriptive statistics for task 1 objective results where *n* is the sample size, $\bar{x}$ is the mean, *s* is the standard deviation, *max* is the maximum value of the sample and *min* is the minimum value of the sample. (*) means significant differences between groups and (a) means significant differences in the post hoc test between the marked groups at the level of significance ($\alpha = .05$). Differences in totals are due to malfunctions while operating capture software.

| Measure | Group | n | $\bar{x}$ | s | max | min |
|---|---|---|---|---|---|---|
| Elapsed seconds (*) | ShExML (a) | 7 | 1,560.1429 | 541.57376 | 2,192 | 782 |
| | YARRRML (a) | 6 | 2,443.8333 | 375.44502 | 2,896 | 1,891 |
| | SPARQL-Generate | 4 | 2,292.7500 | 533.49063 | 2,769 | 1,634 |
| | Total | 17 | 2,044.4118 | 620.68370 | 2,896 | 782 |
| Keystrokes | ShExML | 6 | 1,138.50 | 610.588 | 2,287 | 674 |
| | YARRRML | 4 | 1,187 | 449.649 | 1,795 | 810 |
| | SPARQL-Generate | 3 | 1,125.67 | 121.476 | 1,265 | 1,042 |
| | Total | 13 | 1,150.46 | 457.183 | 2,287 | 674 |
| Left button clicks | ShExML | 6 | 176.50 | 112.169 | 327 | 58 |
| | YARRRML | 4 | 318.75 | 177.989 | 551 | 170 |
| | SPARQL-Generate | 3 | 166 | 78.791 | 254 | 102 |
| | Total | 13 | 217.85 | 138.267 | 551 | 58 |
| Right button clicks | ShExML | 6 | 2.17 | 2.137 | 6 | 0 |
| | YARRRML | 4 | 2.25 | 1.708 | 4 | 0 |
| | SPARQL-Generate | 2 | 4.50 | 2.121 | 6 | 3 |
| | Total | 12 | 2.58 | 2.021 | 6 | 0 |
| Mouse wheel scroll | ShExML | 6 | 148 | 183.737 | 486 | 13 |
| | YARRRML | 4 | 679.25 | 606.711 | 1,404 | 101 |
| | SPARQL-Generate | 3 | 199 | 131.160 | 348 | 101 |
| | Total | 13 | 323.23 | 412.819 | 1,404 | 13 |
| Meters travelled by the mouse | ShExML | 7 | 30.400 | 24.318 | 70.079 | 0 |
| | YARRRML | 6 | 43.454 | 43.144 | 101.767 | 0 |
| | SPARQL-Generate | 4 | 21.220 | 16.526 | 37.680 | 0 |
| | Total | 17 | 32.847 | 30.550 | 101.767 | 0 |
| Completeness percentage (*) | ShExML (a) | 7 | 0.771 | 0.296 | 1 | 0.19 |
| | YARRRML | 6 | 0.323 | 0.366 | 0.82 | 0 |
| | SPARQL-Generate (a) | 4 | 0.02 | 0.04 | 0.08 | 0 |
| | Total | 17 | 0.436 | 0.415 | 1 | 0 |
| Precision (*) | ShExML (a) | 7 | 0.495 | 0.286 | 1 | 0.07 |
| | YARRRML | 6 | 0.131 | 0.160 | 0.38 | 0 |
| | SPARQL-Generate (a) | 4 | 0.005 | 0.01 | 0.02 | 0 |
| | Total | 17 | 0.251 | 0.292 | 1 | 0 |

variability depending on the used tool. This suggests that web interfaces used as online development environments are more or less homogeneous and do not have an impact on the development of the scripts. However, keystrokes variable results should be considered with caution because for SPARQL-Generate the mean of completeness percentages was very low; therefore, achieving a final solution may involve more keystrokes. On the other hand, elapsed seconds, completeness percentage and precision show significant differences between groups which suggest that the selected language has an influence on these variables.

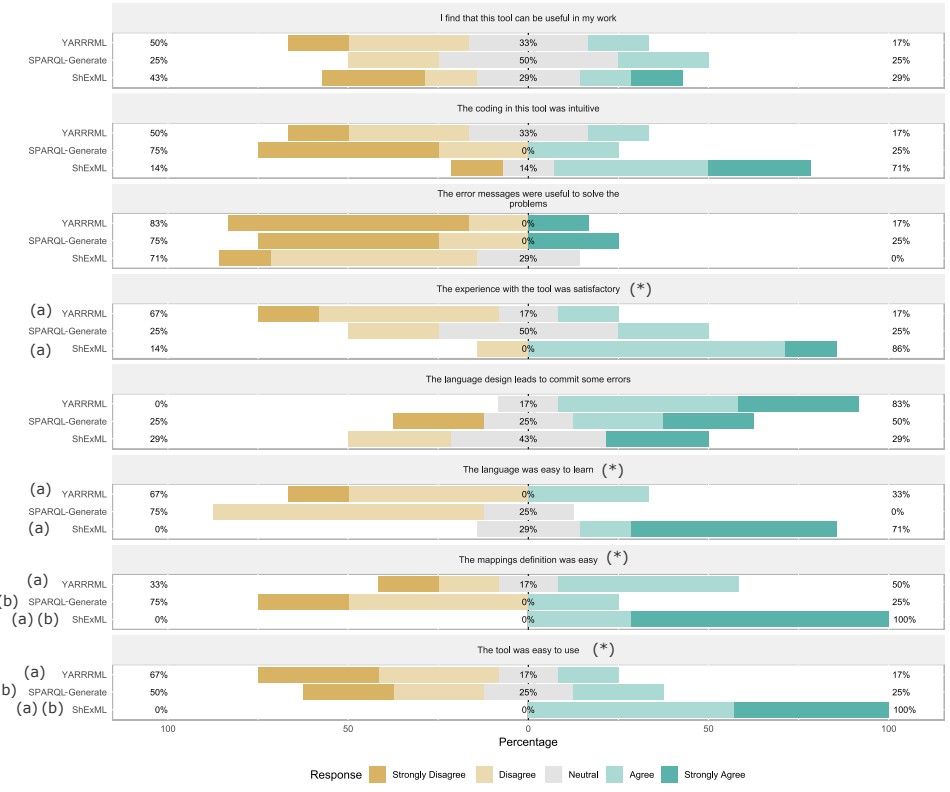

**Figure 1 Task 1 results for Likert scale questionnaire where results are divided into questions and groups.** (*) means significant differences between groups and (a) and (b) means significant differences in the post hoc test between the marked groups at the level of significance α = .05.

Moreover, we can see that elapsed seconds has a medium size effect (ω = .60). Post hoc results show that there are significant differences between ShExML and YARRRML which suggests that YARRRML users tend to need more time than ShExML users for these tests. In the case of comparisons with SPARQL-Generate there are not significant differences which can be due to the small sample size and the low completeness percentage. Differences between ShExML and SPARQL-Generate for completeness percentage and precision suggest that SPARQL-Generate users were not able to achieve working solutions as ShExML users, which have the highest mean on both variables. However, between ShExML and YARRRML groups there were no significant differences which is in line with the great variability of those two variables.

Results of task 2 do not show any significant difference between the ShExML group and the YARRRML group. This can be explained by the low sample size in the YARRRML group where only one individual made this step. However, completeness percentage and precision show us that some students did achieve a correct solution with ShExML, whereas in YARRRML group and in SPARQL-Generate group they did not. This leads to the conclusion that only the ShExML group managed to find a working solution to both

**Table 4   Descriptive statistics for task 2 objective results where *n* is the sample size, $\bar{x}$ is the mean, *s* is the standard deviation, *max* is the maximum value of the sample and *min* is the minimum value of the sample.** Differences in totals are due to malfunctions while operating capture software.

| Measure | Group | *n* | $\bar{x}$ | *s* | *max* | *min* |
|---|---|---|---|---|---|---|
| Elapsed seconds | ShExML | 6 | 325.5 | 328.9248 | 879 | 3 |
| | YARRRML | 1 | 47 | 0 | 47 | 47 |
| | Total | 7 | 285.7143 | 318.1822 | 879 | 3 |
| Keystrokes | ShExML | 5 | 206.40 | 175.832 | 438 | 43 |
| | YARRRML | 1 | 91 | 0 | 91 | 91 |
| | Total | 6 | 187.17 | 164.174 | 438 | 43 |
| Left button clicks | ShExML | 5 | 61.80 | 81.417 | 207 | 16 |
| | YARRRML | 1 | 43 | 0 | 43 | 43 |
| | Total | 6 | 58.67 | 73.225 | 207 | 16 |
| Right button clicks | ShExML | 5 | 0.40 | 0.548 | 1 | 0 |
| | YARRRML | 1 | 0 | 0 | 0 | 0 |
| | Total | 6 | 0.33 | 0.516 | 1 | 0 |
| Mouse wheel scroll | ShExML | 5 | 123.80 | 129.494 | 288 | 0 |
| | YARRRML | 1 | 41 | 0 | 41 | 41 |
| | Total | 6 | 110 | 120.655 | 288 | 0 |
| Meters travelled by the mouse | ShExML | 6 | 9.7629 | 13.8829 | 37.7565 | 0 |
| | YARRRML | 1 | 11.7563 | 0 | 11.7563 | 11.7563 |
| | Total | 7 | 10.0477 | 12.6957 | 37.7565 | 0 |
| Completeness percentage | ShExML | 6 | 0.73 | 0.3904 | 1 | 0 |
| | YARRRML | 1 | 0 | 0 | 0 | 0 |
| | Total | 7 | 0.6257 | 0.4507 | 1 | 0 |
| Precision | ShExML | 6 | 0.4683 | 0.37467 | 1 | 0 |
| | YARRRML | 1 | 0 | 0 | 0 | 0 |
| | Total | 7 | 0.4014 | 0.38512 | 1 | 0 |

proposed tasks. Nevertheless, these conclusions must be validated with bigger experiments to have statistical confidence.

The differences in completeness percentage and precision between ShExML and SPARQL-Generate and also between ShExML and YARRRML in elapsed seconds can lead us to the conclusion that usability on first-time users is improved by using ShExML over the other two languages, which answers RQ1. Moreover, this conclusion is reinforced by the situation that in task 2 neither YARRRML nor SPARQL-Generate users were able to find a solution to this task.

Regarding the subjective analysis, significant differences were found between groups in general satisfaction level, mapping definition easiness easiness of use and learnability (as perceived by the students).

On general satisfaction level significant differences were found between ShExML and YARRRML which indicates that ShExML users were more satisfied with the overall use of the tool respect to the YARRRML users. Differences between SPARQL-Generate users and the two other groups could not be established due to their low completeness percentage and precision rates.

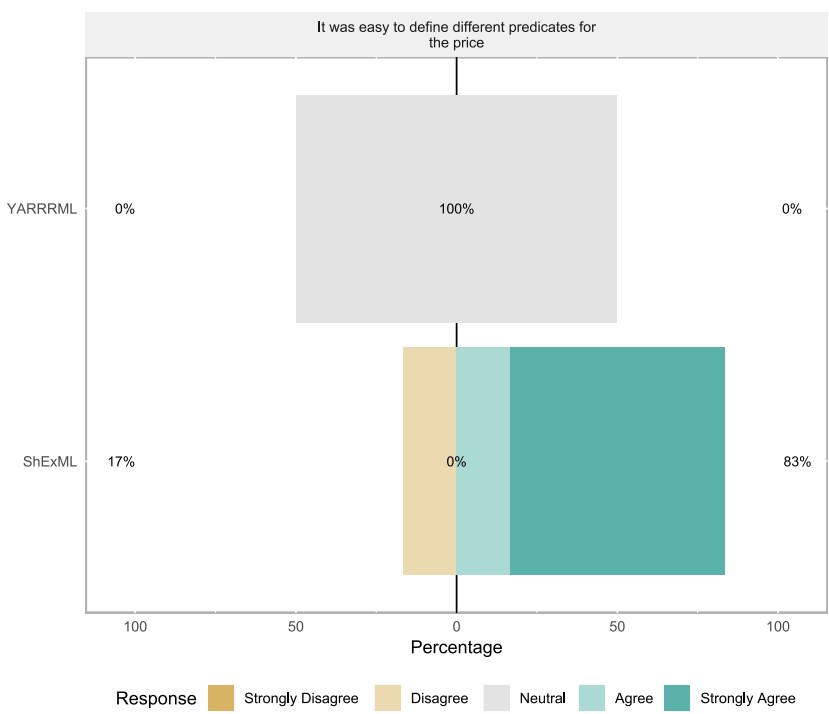

**Figure 2  Task 2 results for Likert scale questionnaire where results are divided into the two groups.**

In the case of easiness of use significant differences were found between ShExML and YARRRML which suggests that ShExML users found this language easier to use than YARRRML users did with their language counterpart. In this case, like in the previous variable, significant differences could not be established between SPARQL-Generate and the two other groups due to low completeness percentage.

In mapping definition easiness differences were established between ShExML group and YARRRML group and between ShExML group and SPARQL-Generate group which indicates that ShExML users found mappings easier to define in ShExML than in the other two languages. We also note that users did not find differences on mapping definition easiness between YARRRML and SPARQL-Generate, this may be because SPARQL-Generate users did not use the whole language.

On learnability significant differences were found between ShExML and SPARQL-Generate and between ShExML and YARRRML which suggests that the users found easier to learn ShExML than the other two languages. However, no significant differences were found between YARRRML and SPARQL-Generate which seems strange due to the difference of verbosity between the two languages.

Differences on subjective analysis between ShExML and YARRRML on general satisfaction level, mapping definition easiness, easiness of use and learnability, and between ShExML and SPARQL-Generate on mapping definition easiness and learnability comes to corroborate what we have elucidated with the objective analysis answering RQ1.

Review of the other variables shows that the users do not see much applicability on the three languages, that the design of the languages leads users to commit some errors during the development of the script and that the error reporting system in the three of them is not very useful to solve the incoming problems.

The feedback received from the users in the error proneness and error reporting usefulness variables determines that these two aspects are the ones that should be improved in the three languages to improve their usability. This comes to answer the RQ3.

For the modifiability variable assessed in task 2, ShExML users tend to rate this feature with high marks whereas the single YARRRML user gave a response of 3 in a 5 point Likert scale which is in line with his/her completeness percentage mark. As with the objective results of task 2, these subjective results should be further validated in future bigger experiments to corroborate these early findings.

## Alignment with features comparison

In the light of the statistical analysis outcome, SPARQL-Generate design has been shown to have a negative impact on first-time users. This led to three users abandoning the task and low completeness scores for the rest of the group. Although having more features in a language is something good and desirable, these results caught attention on how these features should be carefully designed and included in the language in order to improve easiness of use, and thus overall adoption of the tool. In the case of YARRRML language, although it has been designed with human-friendliness in mind, in our experiment it has not reached the expected results in comparison with ShExML. However, it has better results than SPARQL-Generate, suggesting it is less complex to use than that language, but still more complex than ShExML. Nevertheless, it does not seem that supported features could explain the differences between YARRRML and ShExML as the features used on the experiment are more or less equal. Instead other syntax details may be affecting the differences between these two groups such as: the use of keywords that made the language more self explanatory and the modularity used on iterators which reminds of object-oriented programming languages. However, this would require a broader study taking into account programming style background of participants and their own style preferences using techniques like a cognitive complexity architecture (*Hansen, Lumsdaine & Goldstone, 2012*) to identify how each feature and its design is affecting the usability of each specific language.

These results highlight the importance on how features are designed and included in a language. Therefore, SPARQL-Generate with more features and being a highly flexible language tends to have a bad influence on users' usability. Comparing ShExML and YARRRML we see that these differences are smaller than with SPARQL-Generate and that features support does not seem to be the variable affecting YARRRML usability. Thus, we can conclude—and answer the RQ2—that it is not the features supported by a language which affects usability of first-time users but their design.

## CONCLUSIONS AND FUTURE WORK

In this work we have compared the usability of three heterogeneous data mapping languages. The findings of our user study were that better results, and speed on finding this solution, are related to ShExML users whereas SPARQL-Generate users were not able to find any solution under study conditions. In the case of YARRRML users, they performed better than SPARQL-Generate users but worse than ShExML users finding partial solutions to the given problem.

This study is (to our knowledge) the first to explore the topic of usability for first-time users with programming and Linked Data background in these kind of languages. It also reflects the importance that usability has on the accuracy of the encountered solutions and how features should be carefully designed in a language to not impact negatively on its usability.

As future work, bigger experiments should be carried out with an emphasis on programming style background and styles (using cognitive complexity frameworks) to corroborate and expand these early findings. In addition, improving these aspects that were worst rated in the three languages (i.e., error proneness and the error reporting system) would enhance perceived user friendliness.

This work highlights the importance of usability on these kind of languages and how it could affect their adoption.

## ACKNOWLEDGEMENTS

We want to thank the students of the Master's Degree in Web Engineering for their willingness to participate in the experiment described in this work.

### Funding

This work has been funded by the Principality of Asturias through the Severo Ochoa call (grant BP17-29), by the Ministry of Economy, Industry and Competitiveness under the call of ''Programa Estatal de I+D+i Orientada a los Retos de la Sociedad'' (project TIN2017-88877-R), the CPER Nord-Pas de Calais/FEDER DATA Advanced data science and technologies 2015–2020, and the ANR project DataCert ANR-15-CE39-0009. There was no additional external funding received for this study. The funders had no role in study design, data collection and analysis, decision to publish, or preparation of the manuscript.

### Grant Disclosures

The following grant information was disclosed by the authors:
Principality of Asturias through the Severo Ochoa call: BP17-29.
Ministry of Economy, Industry and Competitiveness under the call of ''Programa Estatal de I+D+i Orientada a los Retos de la Sociedad'': TIN2017-88877-R.
The CPER Nord-Pas de Calais/FEDER DATA Advanced data science and technologies 2015-2020.

The ANR project DataCert: ANR-15-CE39-0009.

## Competing Interests

The authors declare there are no competing interests.

## Author Contributions

- Herminio García-González conceived and designed the experiments, performed the experiments, analyzed the data, performed the computation work, prepared figures and/or tables, authored or reviewed drafts of the paper, and approved the final draft.
- Iovka Boneva conceived and designed the experiments, analyzed the data, performed the computation work, prepared figures and/or tables, authored or reviewed drafts of the paper, and approved the final draft.
- Sławek Staworko and Juan Manuel Cueva Lovelle performed the computation work, authored or reviewed drafts of the paper, and approved the final draft.
- José Emilio Labra-Gayo conceived and designed the experiments, performed the experiments, performed the computation work, authored or reviewed drafts of the paper, and approved the final draft.

## Ethics

The following information was supplied relating to ethical approvals (i.e., approving body and any reference numbers):

The University of Oviedo granted ethical approval to carry out the described study.

## Data Availability

Experiment data, supplemental material and raw data are available at GitHub: https://github.com/herminiogg/shexml-paper-2019-data.

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
