# Peer review of "ShExML: improving the usability of heterogeneous data mapping languages for first-time users"

_PeerJ Computer Science, doi:10.7717/peerj-cs.318_

## Round 0.1 · original submission · Major Revisions

· Academic Editor

Major Revisions

Both reviewers have made suggestions on ways your manuscript can be improved for clarity, and also to address omissions or lack of detail in important aspects of the design and evaluations in the study.

I look forward to receiving your revised manuscript.

·

Basic reporting

Summary:
Based on prior research this paper asserts that proposed domain-specific language (DSL) support is needed in the data transformation among many sources (e.g. JSON and XML) into a RDF output. The paper suggests that there are weaknesses/limitations in the usability of DSLs available currently, and so proposes a new language that allows to transform and integrates data from XML and JSON sources into a single RDF output. An experiment evaluation was conducted alongside an expressiveness comparison among three DSLs (YARRRML, SPARQL Generate and ShExML). The experiment results compares the development time and learning curve metrics as well as the effect on usability of these DSLs. The authors conclude that their DSL is superior on usability aspects to the alternatives and it is going in the right direction.

Pros:
+ Good fit with the scope of the PeerJ Computer Science Journal
+ Topic (data integration and transformation) is of practical and research relevance
+ Good chosen DSLs/languages to evaluate the usability of them for data integration and transformation
+ It was good to see the use of prior studies providing a basis for the features one to one transformations and many to one transformations
+ I liked the intention of authors being rigorous in the experiment reported, in the spite of not adequately formalized and referenced.
+ In general, the paper uses a clear language in terms of the quality of expression

Cons:
- The authors are vague on the make-up of the profile of participants, information that is essential to reader understanding the analysis result
- The threat to validity is not presented
- Precision metric could be better formalized
- Improving the quality and format of the listings
- Improving the quality of the Figures using Tikz and Pgfplots packages, some of them have insufficient resolution (Fig.1 and Fig.2). Besides, use the patterns to fill the sample (lines, dots, dashes, ...) instead of colors in the graphs.
- Some tables and figures are overfull the margin limit

Comments:
* The listenings could be configured to highlight the commands and statements, this is worth to know the syntax and the difference among literals, strings, keywords and so on, in order to improve the usability perception of each language under study.
Structure:
* I missed a Related Work section, mention other works that, somehow, are related to the proposal study
* Results section could be a subsection of Methodology
* I missed a Threat to Validity subsection inside Methodology section

Typos:
- 341: our films example > our example films
- 514: a experiment > an experiment

Experimental design

* I missed a Threat to Validity subsection inside Methodology section.
- Creswell, John W and Creswell, J David. 2017. Research design: qualitative, quantitative, and mixed methods approaches. 2. ed. London: Sage Publications.
* The Methodology section should be better improved, I suggest the following structured:
5 Methodology
5.1 Experiment Protocol
5.2 Conduction
5.3 Analysis Result
5.4 Threat to Validity
* Furthermore, the protocol experiment should be better formalized following a rigorous standard and procedures. See the paper:
- Claes Wohlin, Per Runeson, Martin Hst, Magnus C. Ohlsson, Bjrn Regnell, and Anders Wess ln. 2012. Experimentation in Software Engineering. Springer Publishing Company, Incorporated.

Validity of the findings

* In general, the statistical analysis was very well done, showing excellent results.
* It is a pity that the adherence of the number of experiment participants to the second task (Task 2) was low, making statistical analysis difficult.
* For improvements, I suggest that authors for internal cohesion of responses by applying the Cronbach's Alpha analysis, and mode the responses per respondent and per question.
- Mohsen Tavakol and Reg Dennick. Making Sense of Cronbach’s Alpha. International Journal of Medical Education. 2011; 2:53-55 Editorial.
* Precision could be better formalized. For instance using class agreement of the data labels with the positive labels given by the classifier.
- Precision: the number of correctly classified positive examples divided by the number of examples labeled by the system as positive:
- Precision = TP / (TP / FP), where TP: True Positive and FP: False Positive
- Marina Sokolovaa and Guy Lapalmeb A systematic analysis of performance measures for classification tasks. Information Processing & Management, Volume 45, Issue 4, July 2009, Pages 427-437. https://doi.org/10.1016/j.ipm.2009.03.002
* Again, I claim a Threat to Validity section, describing the internal, external, construct and conclusion validities of the controlled experiment.

Additional comments

To assist the author(s) in revising his/her/their manuscript, please separate your remarks into two sections:
(1) Suggestions which would improve the quality of the paper but are not essential for publication.
See the comments and suggestions described in the 1. Basic reporting, 2. Experimental design and 3. Validity of the findings sections.
(2) Changes which must be made before publication
In general, the article is well written, easy to understand and, in the current format, has merits for acceptance in PeerJ. It is recommended to rewrite the Methodology section, including details about experiment protocol and threat to validity. The Background section presents important concepts and examples of DSLs/languages, but the Related Work is unclear and not discussed in depth with appropriate comparison. In other words, the Related Work section is poor, missing a comparison with other similar studies. Besides, the other sections should be improved, following the detailed comments previously exposed. The main point is that scientific value is perceived by the motivation and the achieved results. Empirical studies, mainly experiments, should be conducted following appropriate protocols for evaluation, which is not the case for the reported experiment, it should be better formalized and referenced.

Reviewer 2 ·

Basic reporting

The (English) language could be improved. I indicatively mention a few cases that need improvement but there are more:
* Data integration is the problem of integrating
* This supposed the creation of a new script
* data integration is performed more easily using ShExML (line 57)
* In the way to offer a simpler solution (137)
* Things like iterators, sources, fields, unions and so on (391)
* … but they are more or less similar. (392)

Introduction
* * *
The introduction remains too abstract and doesn’t reflect well the reality. For instance, it is mentioned in the introduction that "more recent proposals allow to directly transform multiple data sources into a single representation” (line 47) or “This is an improvement compared to previous techniques as in principle it allows for faster development and improved maintainability.” (line 49-50). However, the claim is not supported by references, not even examples. This occurs throughout the introduction.

It is also mentioned that “ShExML uses Shape Expressions for defining the desired structure of the output. “ (line 53-54) but it is not clarified what Shape Expressions are.

It is also mentioned that “ShExML is intended for users that have some programming background” but later on it is compared with SPARQL-Generate that is intended for users with Linked Data background. In that case, I am not sure if one can compare the two languages.

“We focus on usability of tools based on a DSL” —> I guess we may agree that ShExML or the other languages are DSL but that’s nowhere specified in the text.

2 BACKGROUND
——————————
The main problem I see with this section is that the languages and the implementations are mixed for all cases but it is more evident in the first subsection that is focus on one to one transformations. Given that the paper is focused on languages, I would expect the background section to be similarly focused on language based solutions and be complemented by implementations if they somehow become relevant. For instance, it is mentioned that “Bizer and Seaborne (2004) present a platform to access relational databases as a virtual RDF store. “ but in practice and what is interesting, considering that this paper looks int languages usability, is the D2R language they propose rather than the D2RQ platform they proposed next to the language.

In the same context, it is hard to identify the main trends. For instance, in the case of XML, different solutions are put one next to the other but where did the different DSL solutions were based on? XML Schema, XQuery, XSLT, others? And even more, what were the advantages and disadvantages of these approaches? This way, a reader can better understand what the issues were and we need yet another solution that is inspired from a complete different basis.

In the end of the background section, I am not sure if I know which are the trends of languages and I am not sure why the languages that were chosen for the comparison are representative. At the moment, the comparison against SPARQL-Generate and YARRRML seems to be a random choice which I am sure it is not but this is not clear based on the current version of the background section.


2.1.2 From JSON to RDF
* * *
How do you se JSON 1.1 as transformation language for JSON?

2.1.3 From CSV to RDF
* * *
I think in this subsection, the notion of CSV and Excel are mixed but when it comes to their transformation to RDF they have quite some differences, so I would recommend to clear this up. One solution would be to distinguish completely the two, the other solution would be to change the subsection’s header to something like “From tables to RDF” and then cover all possible cases of tables. This would, of course, require to consider more than what’s already explored, included databases that are not covered in this subsection.

I think that this subsection does not fully reflect reality overall. CSVW (CSVs), which is DSL based solution but most importantly the only other W3C recommendations for transforming data to RDF, is not mentioned. Besides CSVW, other important contributions, such as Tarql (https://tarql.github.io/) are not mentioned at all. Similar comment goes for the databases section, where significant contributions are not mentioned, such as SML (http://sml.aksw.org/).

2.2.2 Text-based approaches
* * *
I am not confident of the title of this subsection. What is it meant by text-based?

2.2.3 Graphical-based approaches
* * *
I think that the graphical-based approaches is an example of what I mentioned as my first comment for the background section. If the paper is focused on languages usability, the tools are out of scope in my opinion but the two different concepts (languages and tools) are mixed throughout the paper.
If this authors think that this section is relevant, I would recommend to put it in context and make it more representative. I am sure there are more tools to mention.

Experimental design

3 PRESENTATION OF THE LANGUAGES UNDER STUDY
* * *
Throughout the paper it is mentioned that the three “languages” will be compared but YARRRML is not a language but a YAML based syntax for RML as it is earlier mentioned. So what is going to be compared is the YARRRML syntax of RML with ShExML and SPARQL-Generate. The same claim appears in the abstract “In this work we have conducted a usability experiment on three different languages: ShExML (our own language), YARRRML and SPARQL Generate. “

Then if “YARRRML and ShExML are able to compile their mappings to RML” then we are talking about "syntax sugar” on top of RML language but how SPARQL-Generate becomes comparable? YARRRML could support other languages besides RML, eg SPARQL-Generate, which brings me to my next comment.

It is mentioned in the text that “RML is considered as a lower level middle language to compile” (151) and this is the reason it was not chosen. I would agree that RML might not be the best language for a user to edit but why does this hold for RML and not for SPARQL-Generate? What about the other languages that were proposed and mentioned in the background section? Where do we draw the line and a certain language may be considered lower level middle language?

At the same part, the sentence continues “ (RML is) to be used by programmers and data engineers” (152) but in the introduction it is mentioned that the hypothesis targets “first-time users with some programming background” (56) which confuses me because if the hypothesis focuses on users with programming background then RML should be considered no? I think what it is considered as user for the study should be clarified throughout the whole paper.

I am not sure if it is a good practice to keep such long lists of XML, JSON examples followed by examples in each language. It might be better if they are placed in the appendix but this is more a matter of styling. I would suggest to check what the journal recommends.

I am not sure why “ The originality of ShExML, compared to the other two languages studied here, is that the output is defined only once even when several sources are used.” (384-386) only holds for ShExML and not for the other syntaxes that ShExML is compared with.

3.4 Comparing expressiveness of the languages
* * *
First of all, it is not explained what is considered in the paper as “expressiveness”. I would suggest to refer to a definition.

My greatest doubt though about this subsection is the list that is used as criteria for expressiveness, e.g., source and output definition, multiple results, transformations etc.
Why are these good criteria? How are we sure that they are complete?
For instance, it is mentioned that “things” like iterators are common in all three languages, their syntax is different but from an expressiveness point of view these features have no differences. (393) However. The KR2RML language for instance (which is not mentioned in the background section) does not require iterators to achieve the same results.

I think this is a very superficial view of what expressiveness might mean in the case of such languages. First of all, their expressiveness is multidimensional. It depends on their expressiveness with respect to the input data, the expressiveness of the language per se and its expressiveness with respect to the output, in this case RDF. On top there are more expressiveness dependencies. For instance, YARRRML’s expressiveness for data transformation depends on FnO’s expressiveness, whereas SPARQL-Generate’s data transformation expressiveness depends on SPARQL. As far as the language’s expressiveness is concerned, SPARQL-Generate’s expressiveness depends on SPARQL whereas YARRRML’s expressiveness depends on RML which, on its own turn, depends on R2RML’s expressiveness. All three languages’ expressiveness depends on their coverage against the RDF specification. For instance, one can generate RDF lists with SPARQL-Generate but cannot do with “core” RML unless the xR2RML extension is considered (xR2RML is also not considered in the background section). Last, the expressiveness of these languages with respect to the input data depends on the formulations they use to refer to them. For instance, if XQuery is considered by one language and XQuery is more expressive than XPath, then the language that considers XQuery to refer to the input data would be more expressive. So, I think that the list of arguments may be considered as possible comparison parameters but I do not think that they can qualify for fully assessing these syntax and languages expressiveness.

Regarding the expressiveness of the ShExML compared to the other languages, how is the expressivity affected by the different formats that are supported? Because ShExML seems to support less data formats.

I think this section would benefit from a comparison table that summarises the differences.

“YARRRML supports named graphs whereas the other two do not.” Are you sure that SPARQL-Generate does not support named graphs?

“in SPARQL Generate it is possible to also transform to text formats, enabling the transformation to a lot of different formats.” (416) I am not sure I understand this

In the end, I don’t understand how we come to the conclusion that SPARQL-Generate has further expressiveness (432-440)

4 FURTHER SHEXML CONSTRUCTIONS
* * *
It is not clear what each feature generates. I would suggest to include more complete examples where not only the ShExML syntax is mentioned but also the expected output

4.1 Nested iterators
* * *
I am not sure I understand what the purpose is of nested iterators and what is generated if they are used.

5 METHODOLOGY
* * *
But based on what we read in the Results section, in the end we have results from 14 students in the first task and only 7 in the second task.
However, in the results results of 17 students are considered even though they didn’t provide complete results. I would suggest to revise.

It is not clarified what the background of the students is. For instance, if the students took classes of ShExML or even ShEx during the semester, it may be explained why they perform faster with ShExML or why they find this easier compared to SPARQL-Generate if SPARQL was not in the course’s curriculum. Statistics regarding the background knowledge of each student should have been recorded before starting the task and provided in the paper.

Validity of the findings

6 RESULTS
* * *
Given that almost all students but one that filled in the questionnaire for Task 2, only did the task with ShExML, we may only make conclusions for ShExML.
It is not explained what p and ω and H and F. One-way ANOVA, Kruskal-Wallis test and the Gabriel’s criterion are all mentioned but it is not explained why they were chosen and what their parameters prove. For instance, what does it mean that p=0.16 between ShExML and YARRRML and p=0.12 between ShExML and SPARQL-Generate. Why was Gabriel’s criterion used for ShExML and YARRRML and Bonferroni criterion between ShExML and SPARQL-Generate? What about YARRRML and SPARQL-Generate? In the end it is not explicitly explained which syntax is better on what and how the differences are explained.

Why Table 2 has different totals?

For the rest, I am not sure what the number of left and right clicks or mouse scroll and move might indicate. Especially considering if the tool or the language is evaluated.


It is mentioned that "we can see that elapsed seconds has a medium size effect (ω = .60)” (587) but what does this mean?

---

## Round 0.2 · Minor Revisions

· Academic Editor

Minor Revisions

R1 notes the following issues. They also offer a general observation on the way in which languages arise and are perceived which could be useful to acknowledge in your conclusions.

I provide numbered observations and comments on R1's points of revision below.

- clearly distinguish languages that facilitate data mappings and implementations that allow data mappings to be performed.

- Specifically, in section 2.3 - the revised 'related work' section, they suggest you do not clearly report Lefrancois' findings.

1. I have offered some alternative wordings for both these aspects in the PDF.

- Expressiveness is poorly defined, and in this study does not appear to quantitate differences in the evaluated languages.

2. I somewhat agree with R1's concerns and have commented as such at various points in the manuscript (e.g. line 478 and line 762). I recommend you consider whether it is worth carrying out a formal 'expressiveness' analysis of the languages using more widely recognised expressiveness characteristics such as those presented by Batdalov et al. (2017, https://doi.org/10.1515/acss-2016-0012).


They also make the following suggestions:

- A table presenting features common and unique to each language would help this evaluation (such as the one presented by Meester et al.)
3. If time permits, a table would help complement to section 3's comments on the features. I do not consider it a requirement, however.

- more analysis/explanation of the results could be presented.
4. There is certainly opportunity to mine the experiment's results further, but given the small sample sizes this may prove difficult. Eg - as I note in line 655, presentation bias could be alleviated and additional insight gained if records were be kept concerning the questions asked and problems experienced by participants. This would allow subsequent qualitative analysis to identify any significant trends and systematic biases due to the way materials were presented.


In addition, I have noted my own observations and recommendations in the attached PDF - these include a substantial number of suggested rewording for clarity and grammar. I also have the following points of revision:

5. I found the paper still lacks solid theoretical foundation. I suggest you consider citing somewhere in your introduction (and when making revisions, drawing on the principles from) one of the fundamental works on human factors in programming, such as Hanenberg (available at https://courses.cs.washington.edu/courses/cse590n/10au/hanenberg-onward2010.pdf). For cognitive complexity, a concrete framework was offered by Hansen et al. in 2012 https://doi.org/10.1145/2384592.2384596

6. Line 167. It is not clear to me why "RDF syntax" precludes achieving Meester's NF1 'Easy to use by semantic web experts' ? If it is relevant please clarify here.

7. Suggest Section 4 be moved to Supplemental Data - it largely consists of additional language details that are informative, but not actually needed to understand the description of the experiment and its subsequent analysis.

8. Archiving experimental data. The git repository cited in Section 5.1 should be linked to Zenodo (or your preferred data archiving solution) and tagged in order to create a DOI and archival copy, in line with Journal policies.

9. Table 1 attributes 'Learnability' to a questionnaire response. This may be misleading to readers since here learnability refers to the subject's 'perceived learnability' rather than objectively measured learnability (say by having subjects execute one task to learn a particular transformation language, and then evaluate if they can apply the language more effectively one or two days later during another task). Suggest qualifying this aspect when first introduced and when discussed in line 726.

10. I have commented extensively in section 7.2 "Alignment with expressiveness comparison" - if literature can be cited supporting your interpretations I think it is fine to include these reflections, but I again agree with the reviewer that the experiment cannot directly observe the relationship between each language's expressiveness and its usability.

11. I've also suggested revisions to make your conclusions clearer and easier to read. I recommend you request input from a colleague with very good written English to help you reformulate the final sentences!

Reviewer 2 ·

Basic reporting

As far as the background is concerned, I think the authors might not have properly understood my main comment (or I didn’t explain well) with regard to related work. This is even evident in the new subsection that was added (2.3 Related studies). I do not disagree that tools should be mentioned as well if relevant but the languages and tools should not be compared as one and the same. And as long as this holds for the background section, I cannot accept the paper for publishing. I will try to explain the problem with the current text with an example but this is not the only case. That means that the background section should still be reworked to distinguish references to languages and tools and compare only languages with languages and tools with tools. Therefore, the new subsection does not solve the problem. In its current state, apples and oranges are presented and compared as one and the same and this needs to be improved before publishing.

For instance, in the new section it is mentioned “Lefranc¸ois et al. (2017)
tested the performance of SPARQL-Generate in comparison with RML. The results showed that SPARQL-Generate has a better performance when transforming more than 1500 CSV rows in comparison with RML.”

Lefrancois et al. did not test SPARQL-Generate in comparison with RML. Lefrancois et al. compared the SPARQL-Generate implementation which unfortunately happens to have the same name causing confusion with the RMLMapper (not RML). RML is a language and it cannot be compared with a tool. I acknowledge that Lefrancois et al. also phrased it wrongly in the paper but that does not mean we should propagate the wrong phrasing.

The reference to expressiveness definition is not peer reviewed. But more importantly, it does not impact the content of the paper nor explains the features choice. I would suggest to the authors to drop the expressiveness comparison claim and turn it into features comparison or something alike because there is no proof that the chosen features reflect expressiveness. Similarly, the conclusion that SPARQL-Generate is more expressive should be dropped as it is only proven it supports more features but those features do not prove expressiveness.

I would still like to see a comparison table even if this is a yes/no table and I think this is inline with the other reviewer’s comments.

Experimental design

I’d like to thank the authors for clarifying the background of the students. It is now clear that they had background knowledge directly related to two out the three languages that were compared but not the third one. I would suggest to have this explicitly clarified in the text. Relying on an assumption that the students might have background related to YAML remains an assumption. It should have normally declared the background of students with respect to YAML as it is the case with SPARQL and ShEx.

I would still find useful some explanation of the impact of the results. I did not mean that the statistical measures need to be introduced but an explanation of the results would be beneficial for the paper instead of just reporting numbers.

Validity of the findings

no comment

Additional comments

I would like to thank the authors for their detailed answers overall.

A last comment with regard to easiness of use without expectation of reaction:
The fact that certain languages explicitly claim easiness for use while others not does not mean that the others cannot be easy for use and do not deserve to be compared. For instance, RML was the first language of its kind, we may agree it’s not the easiest to use but when introduced the concept of mapping languages for heterogeneous data sources did not even exist, so of course the language’s purpose at that point was not to prove easiness of use.

---

## Round 0.3 · accepted · Accept

· Academic Editor

Accept

I have provided an annotated PDF where I have highlighted several typographic and grammatical errors that should be addressed in the final version of your work.

I note 3 minor revisions and offer suggested solutions in comments in the PDF:

1. Line 492. You describe task 1 as mapping an integration of JSON and XML containing data about books. A little more information here will help the reader understand the relevance of task 2.

2. Line 506: regarding prior knowledge of YAML by subjects.

3. Line 672-675: Strongly suggest the latter part of the sentence (see below) is omitted, or replaced with "in order to improve ease of use, and thus overall adoption of the tool".

latter part of sentence: "in order to not affect variables such as the one measured in this study and, therefore, affect other variables like the adoption of the tool. "

In my opinion it is quite difficult to conceive of a way of designing a language in a way that optimises *observed* human factors without identifying the underlying language features that affect these factors. Your experiment has suggested some design factors that affect easiniess of use, etc, but your data says nothing about how SPARQL-Generate could be redesigned to improve its easiness of use.

Moreover, you recapitulate this idea yourself in line 686: so perhaps this statement can be simply omitted.